# Reverse engineering learned optimizers reveals known and novel mechanisms

## Abstract

Learned optimizers are algorithms that can themselves be trained to solve optimization problems. In contrast to baseline optimizers (such as momentum or Adam) that use simple update rules derived from theoretical principles, learned optimizers use flexible, high-dimensional, nonlinear parameterizations. Although this can lead to better performance in certain settings, their inner workings remain a mystery. How is a learned optimizer able to outperform a well tuned baseline? Has it learned a sophisticated combination of existing optimization techniques, or is it implementing completely new behavior? In this work, we address these questions by careful analysis and visualization of learned optimizers. We study learned optimizers trained from scratch on three disparate tasks, and discover that they have learned interpretable mechanisms, including: momentum, gradient clipping, learning rate schedules, and a new form of learning rate adaptation. Moreover, we show how the dynamics of learned optimizers enables these behaviors. Our results help elucidate the previously murky understanding of how learned optimizers work, and establish tools for interpreting future learned optimizers.

## 1 Introduction

Optimization algorithms underlie nearly all of modern machine learning. A recent thread of research is focused on learning optimization algorithms, by directly parameterizing and training an optimizer on a distribution of tasks. These so-called *learned optimizers* have been shown to outperform baseline optimizers in restricted settings (Andrychowicz et al., 2016; Wichrowska et al., 2017; Lv et al., 2017; Bello et al., 2017; Li & Malik, 2016; Metz et al., 2019; 2020).

Despite improvements in the design, training, and performance of learned optimizers, fundamental questions remain about their behavior. We understand remarkably little about *how* these systems work. Are learned optimizers simply learning a clever combination of known techniques? Or do they learn fundamentally new behaviors that have not yet been proposed in the optimization literature? If they did learn a new optimization technique, how would we know?

Contrast this with existing "hand-designed" optimizers such as momentum (Polyak, 1964), AdaGrad (Duchi et al., 2011), RMSProp (Tieleman & Hinton, 2012), or Adam (Kingma & Ba, 2014). These algorithms are motivated and analyzed via intuitive mechanisms and theoretical principles (such as accumulating update velocity in momentum, or rescaling updates based on gradient magnitudes in RMSProp or Adam). This understanding of underlying mechanisms allows future studies to build on these techniques by highlighting flaws in their operation (Loshchilov & Hutter, 2018), studying convergence (Reddi et al., 2019), and developing deeper knowledge about why key mechanisms work (Zhang et al., 2020). Without analogous understanding of the inner workings of a learned optimizers, it is incredibly difficult to analyze or synthesize their behavior.

In this work, we develop tools for isolating and elucidating mechanisms in nonlinear, high-dimensional learned optimization algorithms (§3). Using these methods we show how learned optimizers utilize both known and novel techniques, across three disparate tasks. In particular, we demonstrate that learned optimizers learn momentum (§4.1), gradient clipping (§4.2), learning rate schedules (§4.3), and a new type of learning rate adaptation (§4.4). Taken together, our work can be seen as part of a new approach to scientifically interpret and understand learned algorithms. We provide code for training and analyzing learned optimizers, as well as the trained weights for the learned optimizers studied here, at `redacted URL`.

## 2 BACKGROUND AND RELATED WORK

We are interested in optimization problems that minimize a loss function ($f$) over parameters ($\boldsymbol{x}$). We focus on first-order optimizers, which at iteration $k$ have access to the gradient $g_i^k \equiv \nabla f(x_i^k)$ and produce an update $\Delta x_i^k$. These are *component-wise* optimizers that are applied to each parameter or component ($x_i$) of the problem in parallel. Standard optimizers used in machine learning (e.g. momentum, Adam) are in this category[1]. Going forward, we use $x$ for the parameter to optimize, $g$ for its gradient, $k$ for the current iteration, and drop the parameter index ($i$) to reduce excess notation.

An optimizer has two parts: the optimizer state ($\boldsymbol{h}$) that stores information about the current problem, and readout weights ($\boldsymbol{w}$) that update parameters given the current state. The optimization algorithm is specified by the initial state, the state transition dynamics, and readout, defined as follows:

$$\boldsymbol{h}^{k+1} = F(\boldsymbol{h}^k, g^k) \tag{1}$$
$$x^{k+1} = x^k + \boldsymbol{w}^T \boldsymbol{h}^{k+1}, \tag{2}$$

where $\boldsymbol{h}$ is the optimizer state, $F$ governs the optimizer state dynamics, and $\boldsymbol{w}$ are the readout weights. *Learned optimizers* are constructed by parameterizing the function $F$, and then learning those parameters along with the readout weights through meta-optimization (detailed in Appendix C.2). *Hand-designed* optimization algorithms, by distinction, specify these functions at the outset.

For example, in momentum, the state is a scalar (known as the velocity) that accumulates a weighted average of recent gradients. For momentum and other hand-designed optimizers, the state variables are low-dimensional, and their dynamics are straightforward. In contrast, learned optimizers have high-dimensional state variables, and the potential for rich, nonlinear dynamics. As these systems learn complex behaviors, it has historically been difficult to extract simple, intuitive descriptions of the behavior of a learned optimizer.

Our work is heavily inspired by recent work using neural networks to parameterize optimizers. Andrychowicz et al. (2016) originally showed promising results on this front, with additional studies improving robustness (Wichrowska et al., 2017; Lv et al., 2017), meta-training (Metz et al., 2019), and generalization (Metz et al., 2020) of learned optimizers.

We also build on recent work on reverse engineering dynamical systems. Sussillo & Barak (2013) showed how linear approximations to nonlinear dynamical systems can yield insight into the algorithms used by these networks. More recently, these techniques have been applied to understand trained RNNs in a variety of domains, from natural langauge processing (Maheswaranathan et al., 2019a; Maheswaranathan & Sussillo, 2020) to neuroscience (Schaeffer et al., 2020). Additional work on treating RNNs as dynamical systems has led to insights into their computational capabilities (Jordan et al., 2019; Krishnamurthy et al., 2020; Can et al., 2020).

## 3 METHODS

### 3.1 TRAINING LEARNED OPTIMIZERS

We parametrize the learned optimizer with a recurrent neural network (RNN), similar to Andrychowicz et al. (2016). Specifically, we use a gated recurrent unit (GRU) (Cho et al., 2014) with 256 units. The only input to the optimizer is the gradient. The RNN is trained by minimizing a meta-objective, which we define as the average training loss when optimizing a target problem. See Appendix C.2 for details about the optimizer architecture and meta-training procedures.

We trained these learned optimizers on each of three tasks. These tasks were selected because they are fast to train (particularly important for meta-optimization) and covered a range of loss surfaces (convex and non-convex, low- and high-dimensional):

**Convex, quadratic:** The first task consists of random linear regression problems $f(\boldsymbol{x}) = \frac{1}{2}\|\boldsymbol{Ax} - \boldsymbol{b}\|_2^2$, where $\boldsymbol{A}$ and $\boldsymbol{b}$ are randomly sampled. Much of our theoretical understanding of the behavior of optimization algorithms is derived using quadratic functions, in part because they have a constant

---

[1] Notable exceptions include quasi-Newton methods such as L-BFGS (Nocedal & Wright, 2006) or K-FAC (Martens & Grosse, 2015).

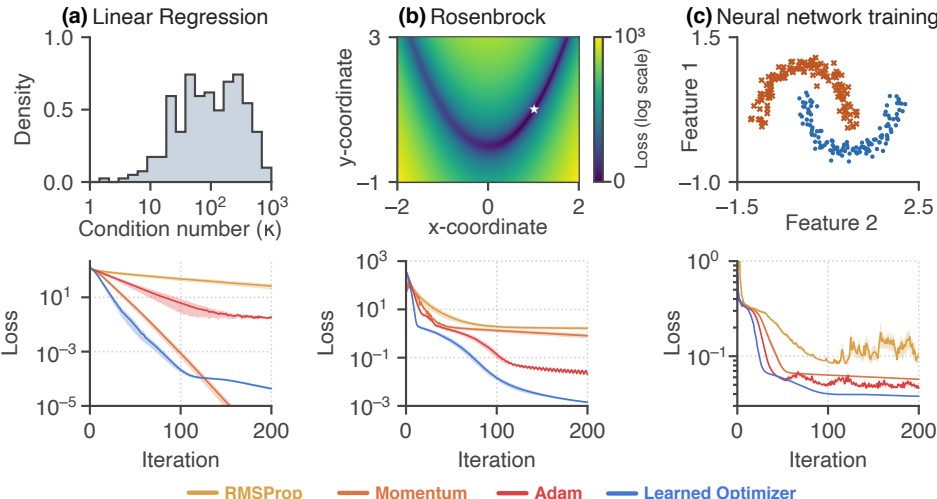

Figure 1: Learned optimizers outperform well tuned baselines on three different tasks: **(a)** linear regression, **(b)** the Rosenbrock function, and **(c)** training a neural network on the two moons dataset. *Upper row*: Task schematics (described in §4). *Bottom row*: Optimizer performance, shown as loss curves (mean ± std. error over 128 random seeds) for different optimizers: momentum (orange), RMSProp (yellow), Adam (red) and a learned optimizer (blue).

Hessian ($A^T A$) over the entire parameter space. The choice of how to sample the problem data $A$ and $b$ will generate a particular distribution of Hessians and condition numbers. The distribution of condition numbers for our task distribution is shown in Figure 1a.

**Non-convex, low-dimensional:** The second task is minimizing the Rosenbrock function (Rosenbrock, 1960), a commonly used test function for optimization. It is a non-convex function which contains a curved valley and a single global minimum. The function is defined over two parameters ($x$ and $y$) as $f(x,y) = (1-x)^2 + 100(y-x^2)^2$. The distribution of problems for this task consists of the same loss function with different initializations sampled uniformly over a grid. The rosenbrock loss surface is shown in Figure 1b, on a log scale to highlight the curved valley. The grid used to sample initializations is the same as the grid shown in the figure; the x-coordinate is sampled uniformly from (-2, 2) and the y-coordinate is sampled uniformly from (-1, 3).

**Non-convex, high-dimensional:** The third task involves training a neural network to classify a toy dataset, the two moons dataset (Figure 1c). As the data are not linearly separable, a nonlinear classifier is required to solve the task. The optimization problem is to train the weights of a three hidden layer fully connected neural network, with 64 units per layer and tanh nonlinearities. The distribution of problems involves sampling the initial weights of the network.

On each task, we additionally tuned three baseline optimizers (momentum, RMSProp, and Adam). We selected the hyperparameters for each problem out of 2500 samples randomly drawn from a grid. Details about the exact grid ranges used for each task are in Appendix C.3.

Figure 1 (bottom row) compares the performance of the learned optimizer (blue) to baseline optimizers (red, yellow, and orange), on each of the three tasks described above. Across all three tasks, the learned optimizer outperforms the baseline optimizers on the meta-objective[2] (Appendix Fig. 9).

### 3.2   PARAMETER UPDATE FUNCTION VISUALIZATIONS

First, we introduce a visualization tool to get a handle on what an optimizer is doing. Any optimizer, at a particular state, can be viewed as a scalar function that takes in a gradient ($g$) and returns a

---

[2]As the meta-objective is the average training loss during an optimization run, it naturally penalizes the training curve earlier in training (when loss values are large). This explains the discrepancy in the curves for linear regression (Fig. 1a, bottom) where momentum continues to decrease the loss late in training. Despite this, the learned optimizer has an overall smaller meta-objective due to having lower loss at earlier iterations.

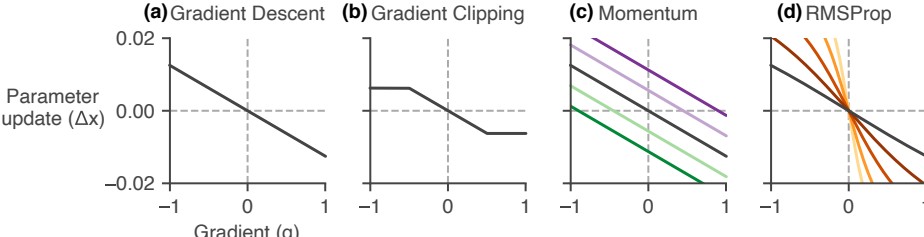

Figure 2: Visualizing optimizer behavior with update functions (see §3.2 for details) for different commonly used optimization techniques. **(a)** Gradient descent is a (stateless) linear function, whose slope is the learning rate. **(b)** Gradient clipping saturates the update, beyond a threshold. **(c)** Momentum introduces a vertical offset depending on the accumulated velocity (colors indicate different values of the accumulated momentum). **(d)** RMSProp changes the slope (effective learning rate) of the update (colors denote changes in the state variable, the accumulated squared gradient).

change in the parameter ($\Delta x$). We refer to this as the optimizer *update function*. Mathematically, the update function is computed as the the state update projected onto the readout, $\Delta x = \boldsymbol{w}^T F(\boldsymbol{h}, g)$, following equations (1) and (2). In addition, the slope of this function with respect to the input gradient $\left( \frac{\partial \Delta x}{\partial g} \right)$ can be thought of as the *effective learning rate* at a particular state[3]. We will use both the overall update function and the effective learning rate to understand optimizer behavior.

It is instructive to visualize these update functions for commonly used optimizers (Figure 2). For gradient descent, the update ($\Delta x = -\alpha g$) is stateless and is always a fixed linear function whose slope is the learning rate, $\alpha$ (Fig. 2a). Gradient clipping is also stateless, but is a saturating function of the gradient (Fig. 2b). For momentum, the update is $\Delta x = -\alpha(v + \beta g)$, where $v$ denotes the momentum state (velocity) and $\beta$ is the momentum hyperparameter. The velocity adds an offset to the update function (Fig. 2c). For adaptive optimizers such as RMSProp, the state variable changes the slope, or effective learning rate, within the linear region of the update function (Fig. 2d). As the optimizer picks up positive (or negative) momentum, the curve shifts downward (or upward), thus incorporating a bias to reduce (or increase) the parameter.

Now, what about learned optimizers, or optimizers with much more complicated or high-dimensional state variables? One advantage of update functions is that, as scalar functions, they can be easily visualized and compared to the known methods in Figure 2. Whether or not the underlying hidden states are interpretable, for a given learned optimizer, remains to be seen.

## 3.3   A DYNAMICAL SYSTEMS PERSPECTIVE

We study the behavior of optimizers by treating them as dynamical systems. This perspective has yielded a number of intuitive and theoretical insights (Su et al., 2014; Wilson et al., 2016; Shi et al., 2019). In order to understand the dynamics of a learned optimizer, we approximate the nonlinear dynamical system via linearized approximations (Strogatz, 2018).

These linear approximations hold near *fixed points* of the dynamics. Fixed points are points in the state space of the optimizer, where — as long as input gradients do not perturb it — the system does not move. That is, an approximate fixed point $\boldsymbol{h}^*$ satisfies the following: $\boldsymbol{h}^* \approx F(\boldsymbol{h}^*, g^*)$, for a particular input $g^*$.

We can numerically find approximate fixed points (Sussillo & Barak, 2013; Maheswaranathan et al., 2019b), by solving an optimization problem where we find points ($\boldsymbol{h}$) that minimize the following loss: $\frac{1}{2}\|F(\boldsymbol{h}, g^*) - \boldsymbol{h}\|_2^2$. The solutions to this problem (there may be many) are the approximate fixed points of the system $F$ for a given input, $g^*$. There may be different fixed points for different values of the input ($g$). First we will analyze fixed points when $g^* = 0$ (§4.1), and then later introduce additional behavior that occurs as $g^*$ varies (§4.4).

---

[3]We compute this slope at $g = 0$. We find that the update function is always linear in the middle with saturation at the extremes, thus the slope at zero is a good summary of the effective learning rate.

## 4 RESULTS

We discovered four mechanisms in the learned optimizers responsible for their superior performance. In the following sections, we go through each in detail, showing how it is implemented. For the behaviors that are task dependent, we highlight how they vary across tasks.

### 4.1 MOMENTUM

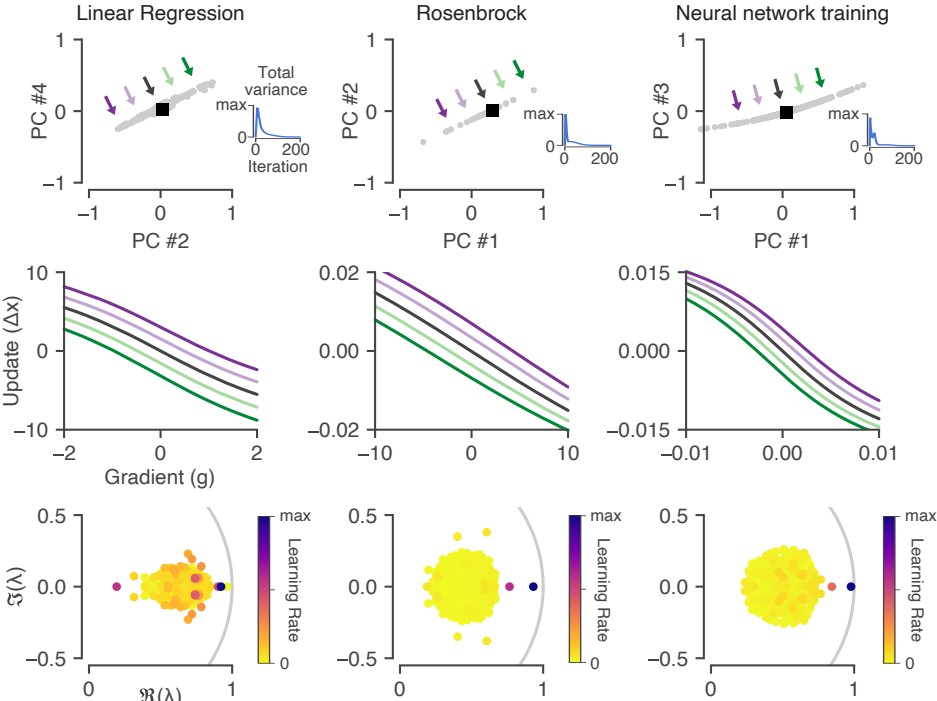

Figure 3: Momentum in learned optimizers. Each column shows the same phenomena, but for optimizers trained on different tasks. **Top row**: Projection of the optimizer state around a convergence point (black square). *Inset:* the total variance of the optimizer states over test problems goes to zero as the trajectories converge. **Middle row**: visualization of the update functions (§3.2) along the slow mode of the dynamics (colored lines correspond to arrows in (a)). Along this dimension, the effect on the system is to induce an offset in the update, just as in classical momentum (cf. Fig. 2c). **Bottom row**: Eigenvalues of the linearized optimizer dynamics at the convergence fixed point (black square in (a)) plotted in the complex plane. The eigenvalue magnitudes are momentum timescales, and the color indicates the corresponding learning rate. See §4.1 for details.

We discovered that learned optimizers implement momentum using approximate linear dynamics (Figure 3). First, we found that each optimizer converges to a single global fixed point of the dynamics. We can see this as the total variance of hidden states across test problems goes to zero as the optimizer is run (inset in Fig. 3). The top row of Fig. 3 is a projection[4] of the hidden state space, showing the convergence fixed point (black square). Around this fixed point, the dynamics are organized along a line (gray circles). Shifting the hidden state along this line (indicated by colored arrows) induces a corresponding shift in the update function (middle row of Fig. 3), similar to what is observed in classical momentum (cf. Fig. 2c).

At a fixed point, we can linearly approximate the nonlinear dynamics of the optimizer using the Jacobian of the state update. This Jacobian is a matrix with $N$ eigenvalues and eigenvectors. Writing

---

[4]We use principal components analysis (PCA) to project the high-dimensional hidden state into 2D. Depending on the task, we found that different mechanisms would correspond to different principal components (hence the different numbers on the x- and y- axes of the top row of Fig. 3).

the update in these coordinates allows us to rewrite the learned optimizer as a momentum algorithm (see Appendix B), albeit with $N$ timescales instead of just one. The magnitude of the eigenvalues are exactly momentum timescales, each with a corresponding learning rate. Note that this type of optimizer has been previously proposed as *aggregated momentum* by Lucas et al. (2018).

We find that learned optimizers use a single mode to implement momentum. The bottom row of Fig. 3 shows the eigenvalues (computed at the convergence fixed point) in the complex plane, colored by that mode's learning rate (see Appendix B for how these quantities are computed). This reveals a single dominant eigenmode (colored in purple), whose eigenvector corresponds to the momentum direction and whose eigenvalue is the corresponding momentum timescale.

While we analyze the best performing learned optimizers in the main text, we did find a learned optimizer on the linear regression task that had slightly worse performance but strongly resembled classical momentum; in fact, this optimizer recovered the optimal momentum parameters for the particular task distribution. We analyze this optimizer in Appendix A as it is instructive for understanding the momentum mechanism.

## 4.2 GRADIENT CLIPPING

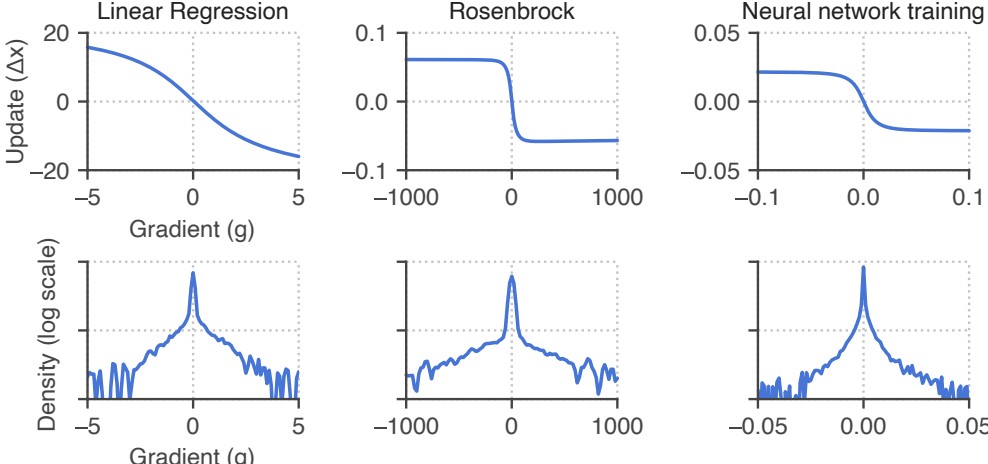

Figure 4: Gradient clipping in a learned optimizer. **Top row**: The update function computed at the initial state saturates for large gradient magnitudes. The effect of this is similar to that of gradient clipping (cf. Fig. 2b). **Bottom row**: the empirical density of encountered gradients for each task (note the different ranges along the x-axes). Depending on the problem, the learned optimizer can tune its update function so that most gradients are in the linear portion of the function, and thus not use gradient clipping (seen in the linear regression task, left column) or can potentially use more of the saturating region (seen in the Rosenbrock task, middle column).

In standard gradient descent, the parameter update is a linear function of the gradient. Gradient clipping (Pascanu et al., 2013) instead modifies the update to be a saturating function (Fig. 2b).

We find that learned optimizers also use saturating update functions as the gradient magnitude increases, thus learning a soft form of gradient clipping (Figure 4). Although we show the saturation for a particular optimize state (the initial state, top row of Fig. 4), we find that these saturating thresholds are consistent throughout the state space.

The strength of the clipping effect depends on the training task. We can see this by comparing the update function to the distribution of gradients encountered for a given task (bottom row of Fig. 4). For some problems, such as linear regression, the learned optimizer largely stays within the linear region of the update function (Fig. 4, left column). For others, such as the Rosenbrock problem (Fig. 4, right column), the optimizer utilizes more of the saturating part of the update function.

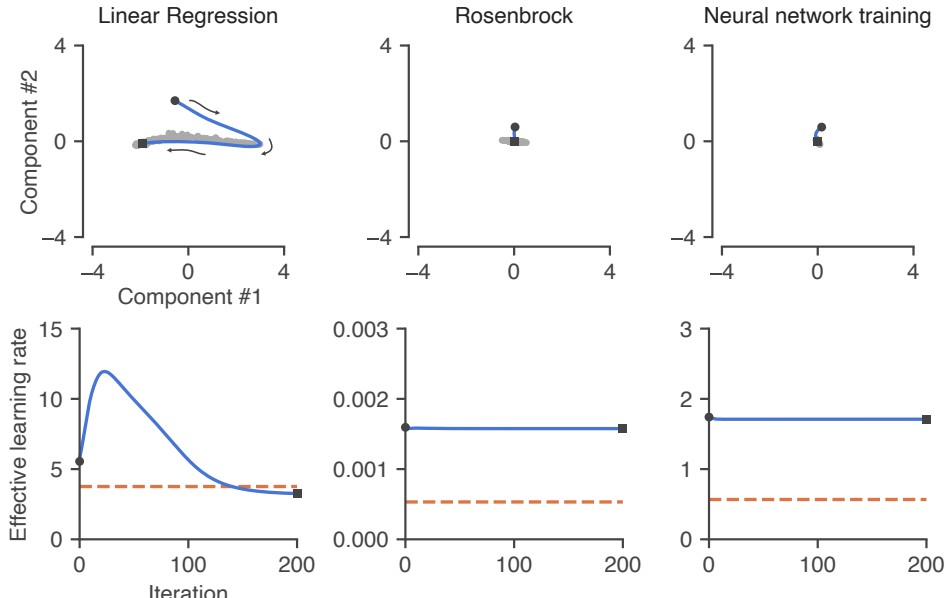

Figure 5: Learning rate schedules mediated by autonomous dynamics. **Top row**: Low-dimensional projection of the dynamics of the learned optimizer in response to zero gradients (no input). These autonomous dynamics allow the system to learn a learning rate schedule (see §4.3). **Bottom row**: Effective learning rate (measured as the slope of the update function) as a function of iteration during the autonomous trajectories in the top row. We only observe a clear learning rate schedule in the linear regression task (left column), which includes both a warm-up and decay. For context, dashed lines indicate the best (tuned) learning rate for momentum.

### 4.3 LEARNING RATE SCHEDULES

Practitioners often tune a learning rate schedule, that is, a learning rate that varies per iteration. Originally motivated for use with stochastic gradients to guarantee convergence to a fixed point (Robbins & Monro, 1951), schedules are now used more broadly (Schaul et al., 2013; Smith et al., 2017; Ge et al., 2019; Choi et al., 2019). These schedules are often a decaying function of the iteration — meaning the learning rate goes down as optimization progresses — although Goyal et al. (2017) use an additional (increasing) warm-up period, and even more exotic schedules have also been proposed (Loshchilov & Hutter, 2016; Smith, 2017; Li & Arora, 2019).

We discovered that learned optimizers can implement a schedule using *autonomous* — that is, not input driven — dynamics. By moving the initial state (which are trainable parameters) away from the convergence fixed point, then even in the absence of input, autonomous dynamics will encode a particular trajectory as a function of the iteration as the system relaxes to the fixed point. Furthermore, this autonomous trajectory evolves in a subspace orthogonal to the readout weights used to update the parameters. This ensures that the autonomous dynamics themselves do not induce changes in the parameters, but only change the effective learning rate.

For the linear regression task, we found a 2D subspace[5] where the autonomous dynamics occur (Figure 5), driving the system from the initial state (black circle) to the final convergence point (black square). The shaded gray points in the top row of Fig. 5 are slow points of the dynamics (Sussillo & Barak, 2013), which shape the trajectory.

By computing the effective learning rate (slope of the update function) of the system along the autonomous trajectory, we can study the effect of these dynamics. We find that for the linear regression task (left column of Fig. 5), the system has learned to initially increase the learning rate over

---

[5]We found this subspace by looking for dimensions that maximized the variation of the autonomous trajectory; this subspace is different from the low-dimensional projection used in Figures 3 and 6.

the course of 25 iterations, followed by a roughly linear decay. We find that the learned optimizer trained on the other tasks does not learn to use a learning rate schedule.

## 4.4 LEARNING RATE ADAPTATION

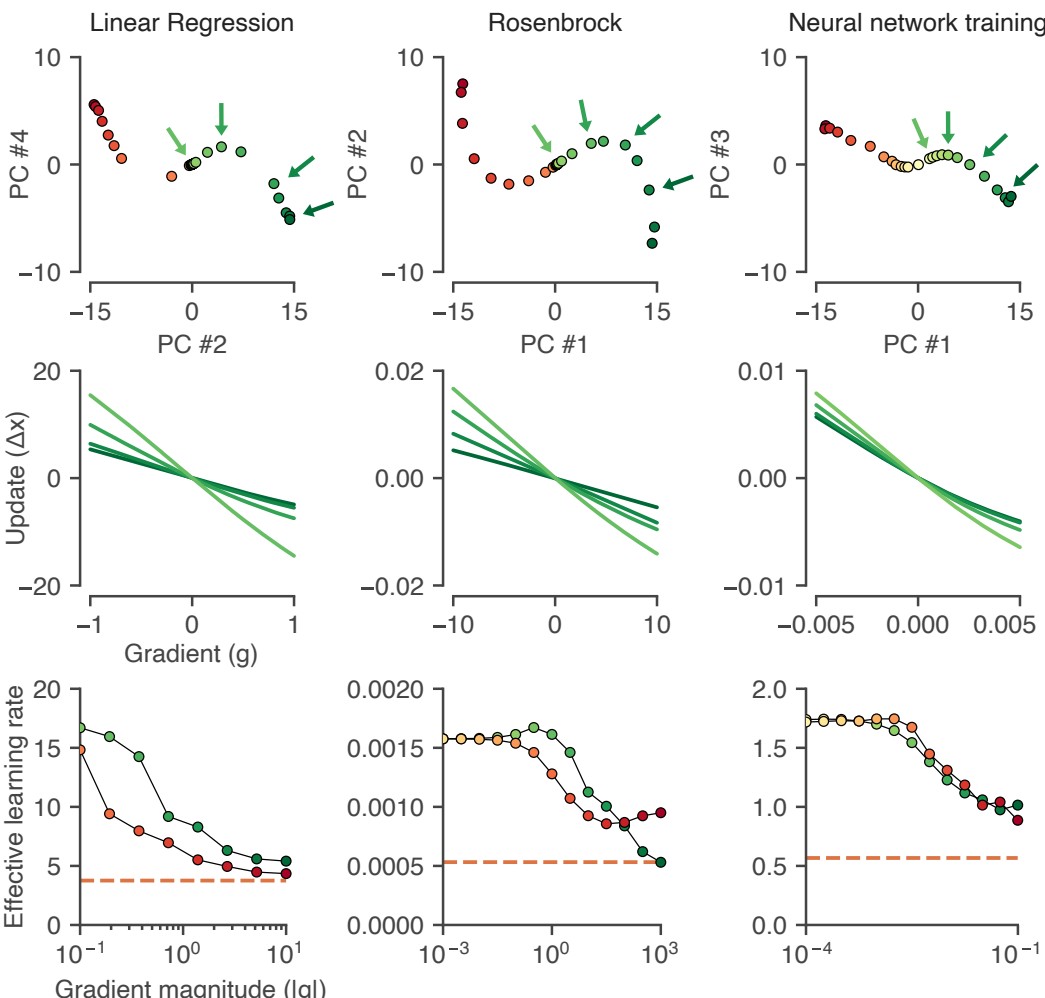

Figure 6: Learning rate adaptation in learned optimizers. **Top row**: Approximate fixed points (colored circles) of the dynamics computed for different gradients reveal an S-curve structure. **Middle row**: Update functions (§3.2) computed at different points along the S-curve (corresponding to arrows from the top row). The effect of moving towards the edge of the S-curve is to make the update function more shallow (thus have a smaller effective learning rate, cf. Fig. 2d). The effect is similar along both arms; only one arm is shown for clarity. **Bottom row**: Summary plot showing the effective learning rate along each arm of the S-curve, for negative (red) and positive (green) gradients. The overall effect is to reduce learning rates when the gradient magnitude is large.

The final mechanism we discovered is a type of learning rate adaptation. The effect of this mechanism is to decrease the learning rate of the optimizer when large gradients are encountered, similar to adaptive learning rate methods such as Adagrad or RMSProp. However, the mechanism that enables this behavior in learned optimizers is novel.

To understand how momentum is implemented by learned optimizers, we studied the linear dynamics of the optimizer near a fixed point (§4.1). That fixed point was found numerically (§3.3) by searching for points $\boldsymbol{h}^*$ that satisfy $\boldsymbol{h}^* \approx F(\boldsymbol{h}^*, g^*)$, where we hold the input (gradient) fixed at zero ($g^* = 0$). To understand learning rate adaptation, we need to study the dynamics around fixed points with non-zero input. We find these fixed points by setting $g^*$ to a fixed non-zero value.

We sweep the value of $g^*$ over the range of gradients encountered for a particular task. For each value, we find a single corresponding fixed point. These fixed points are arranged in an S-curve, shown in the top row of Figure 6. The color of each point corresponds to the value of $g^*$ used to find that fixed point. One arm of this curve corresponds to negative gradients (red), while the other corresponds to positive gradients (green). The tails of the S-curve correspond to the largest magnitude gradients encountered by the optimizer, and the central spine of the S-curve contains the final convergence point[6].

These fixed points are all attractors, meaning that if we held the gradient fixed at a particular value, the hidden state dynamics would converge to that corresponding fixed point. In reality, the input (gradient) to the optimizer is constantly changing, but if a large (positive or negative) gradient is seen for a number of timesteps, the state will be attracted to the tails of the S-curve. As the gradient goes to zero, the system converges to the final convergence point in the central spine.

What is the functional benefit of these additional dynamics? To understand this, we visualize the update function corresponding to different points along the S-curve (middle row of Fig. 6). The curves are shown for just one arm of the S-curve (green, corresponding to positive gradients) for visibility, but the effect is the symmetric across the other arm as well. We see that as we move along the tail of the S-curve (corresponding to large gradients) the slope of the update function becomes more shallow, thus the effect is to decrease the effective learning rate.

The changing learning rate along both arms of the S-curve are shown in the bottom row of Fig. 6, for positive (green) and negative (red) gradients, plotted against the magnitude of the gradient on a log scale. This allows the system to increase its learning rate for smaller gradient magnitudes. For context, the best tuned learning rate for classical momentum for each task is shown as a dashed line.

## 5 DISCUSSION

In this work, we trained learned optimizers on three different optimization tasks, and then studied their behavior. We discovered that learned optimizers learn a plethora of intuitive mechanisms: momentum, gradient clipping, schedules, and learning rate adaptation. While the coarse behaviors are qualitatively similar across different tasks, the mechanisms are tuned for particular tasks.

While we have isolated individual mechanisms, we still lack a holistic picture of how a learned optimizer stitches these mechanisms together. One may be able to extract or distill a compressed optimizer from these mechanisms, perhaps using data-driven techniques (Brunton et al., 2016; Champion et al., 2019) or symbolic regression (Cranmer et al., 2020).

Finally, we are left wondering *why* the learned optimizers we have studied learned these particular mechanisms. Presumably, they have to do with properties of the tasks used to train the optimizer. What are these quantitative properties (e.g. curvature, convexity, or something else)? Understanding these relationships would allow us to take learned optimizers trained in one setting, and know when and how to apply them to new problems.

Previously, not much was known about how learned optimizers worked. The analysis presented here demonstrates that learned optimizers are capable of learning a number of interesting optimization phenomena. The methods we have developed (update functions and visualization of state dynamics) should be part of a growing toolbox we can use to extract insight from the high-dimensional nonlinear dynamics of learned optimizers, and meta-learned algorithms more generally.

---

[6]The top row of Figure 6 uses the same projection as the top row of Figure 3, just zoomed out.

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

## A   A LEARNED OPTIMIZER THAT RECOVERS MOMENTUM

When training learned optimizers on the linear regression tasks, we noticed that we could train a learned optimizer that seemed to strongly mimic momentum, both in terms of behavior and performance. With additional training, the learned optimizer would eventually start to outperform momentum (Figure 1a). We highlight this latter, better performing optimizer in the main text. However, it is still instructive to go through the analysis for the learned optimizer that mimics momentum. This example in particular clearly demonstrates the connections between eigenvalues, momentum, and dynamics.

The learned optimizer that performs as well as momentum learns to mimic linear dynamics (we also used a GRU for this optimizer). That is, the dynamics of the nonlinear optimizer could be very well approximated using a linearization computed at the convergence point. This linearization is shown in Figure 7. We find a single mode pops out of the bulk of eigenvalues (Fig. 7a). Additionally, if we plot these eigenvalue magnitudes, which are the momentum time scales, against the corresponding extracted learning rate of each mode, as discussed below in Appendix B), we see that this mode also has a large learning rate compared to the bulk (top right blue circle in Fig. 7b). Moreover, the extracted momentum timescale and learning rate for this mode essentially exactly match the best tuned hyperparameters (gold star in Fig. 7b) from tuning the momentum algorithm directly, which can also be derived from theory.

Finally, if we extract and run just the dynamics along this particular mode, we see that it matches the behavior of the full, nonlinear optimizer almost exactly (Fig. 7c). This suggests that in this scenario, the learned optimizer has simply learned the single mechanism of momentum. Moreover, the learned optimizer has encoded the best hyperparameters for this particular task distribution in its dynamics. Our analysis shows how to separate the overall mechanism (linear dynamics along eigenmodes) from the particular hyperparameters of that mechanism (the specific learning rate and momentum timescale).

## B   LINEARIZED OPTIMIZERS AND AGGREGATED MOMENTUM

In this section, we elaborate on the connections between linearized optimizers and momentum with multiple timescales. We begin with our definition of an optimizer, equations (1) and (2) in the main text:

$$\begin{aligned} \boldsymbol{h}^{k+1} &= F(\boldsymbol{h}^k, g^k) \\ x^{k+1} &= x^k + \boldsymbol{w}^T \boldsymbol{h}^{k+1}, \end{aligned}$$

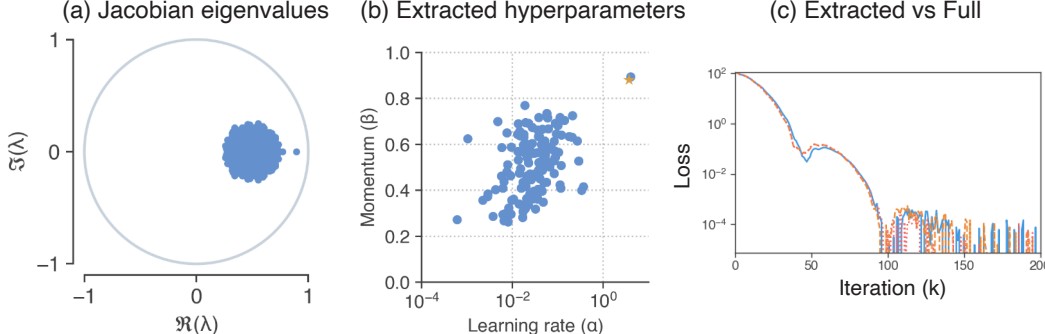

Figure 7: A learned optimizer that recovers momentum on the linear regression task. **(a)** Eigenvalues of the Jacobian of the optimizer dynamics evaluated at the convergence fixed point. There is a single eigenmode that has separated from the bulk. **(b)** Another way of visualizing eigenvalues is by translating them into optimization parameters (learning rates and momentum timescales), as described in Appendix B. When we do this for this particular optimizer, we see that the slow eigenvalue (momentum timescale closest to one) also has a large learning rate. These specific hyperparameters match the best tuned momentum hyperparametrs for this task distribution (gold star). **(c)** When we extract and run just the dynamics along this single mode (orange dashed line), we see that this reduced optimizer matches the full, nonlinear optimizer (solid line) almost exactly.

where $\boldsymbol{h}$ is the optimizer state, $g$ is the gradient, $x$ is the parameter being optimized, and $k$ is the current iteration. Note that since this is a component-wise optimizer, it is applied to each parameter $(x_i)$ of the target problem in parallel; therefore we drop the index $(i)$ to reduce notation.

Near a fixed point of the dynamics, we approximate the recurrent dynamics with a linear approximation. The *linearized* state update can be expressed as:

$$F(\boldsymbol{h}^k, g^k) \approx \boldsymbol{h}^* + \frac{\partial F}{\partial \boldsymbol{h}}\left(\boldsymbol{h}^k - \boldsymbol{h}^*\right) + \frac{\partial F}{\partial g}g^k, \tag{3}$$

where $\boldsymbol{h}^*$ is a fixed point of the dynamics, $\frac{\partial F}{\partial \boldsymbol{h}}$ is a square matrix known as the Jacobian, and $\frac{\partial F}{\partial g}$ is a vector that controls how the scalar gradient enters the system. Both of these latter two quantities are evaluated at the fixed point, $\boldsymbol{h}^*$, and $g^* = 0$.

For a linear dynamical system, as we have now, the dynamics decouple along eigenmodes of the system. We can see this by rewriting the state in terms of the left eigenvectors of the Jacobian matrix. Let $\boldsymbol{v} = \boldsymbol{U}^T \boldsymbol{h}$ denote the transformed coordinates, in the left eigenvector basis $\boldsymbol{U}$ (the columns of $\boldsymbol{U}$ are left eigenvectors of the matrix $\frac{\partial F}{\partial \boldsymbol{h}}$). In terms of these coordinates, we have:

$$\boldsymbol{v}^{k+1} = \boldsymbol{v}^* + \mathbf{B}\left(\boldsymbol{v}^k + \boldsymbol{v}^*\right) + \boldsymbol{a}g^k, \tag{4}$$

where $\mathbf{B}$ is a diagonal matrix containing the eigenvalues of the Jacobian, and $\boldsymbol{a}$ is a vector obtained by projecting the vector that multiplies the input $\left(\frac{\partial F}{\partial g}\right)$ from eqn. (3) onto the left eigenvector basis.

If we have an $N$-dimensional state vector $\boldsymbol{h}$, then eqn. (4) defines $N$ independent (decoupled) scalar equations that govern the evolution of the dynamics along each eigenvector: $v_j^{k+1} = v_j^* + \beta_j\left(v_j^k + v_j^*\right) + \alpha_j g^k$, where we use $\beta_j$ to denote the $j^{\text{th}}$ eigenvalue and $\alpha_j$ is the $j^{\text{th}}$ component of $\boldsymbol{a}$ in eqn. (4). Collecting constants yields the following simplified update:

$$v_j^{k+1} = \beta_j v_j^k + \alpha_j g + \text{const.}, \tag{5}$$

which is exactly equal to the momentum update ($v^{k+1} = \beta v^k + \alpha g^k$), up to a (fixed) additive constant. The main difference between momentum and the linearized momentum in eqn. (5) is that we now have $N$ different momentum timescales. Again these timescales are exactly the eigenvalues of the Jacobian matrix from above. Moreover, we also have a way of extracting the corresponding learning rate associated with eigenmode $j$, as $\alpha_j$. This particular optimizer (momentum with multiple timescales) has been proposed under the name *aggregated momentum* by Lucas et al. (2018).

Taking a step back, we have drawn connections between a linearized approximation of a nonlinear optimizer, and a form of momentum with multiple timescales. What this now allows us to do is interpret the behavior of learned optimizers near fixed points through this new lens. In particular, we have a way of translating the parameters of a dynamical system (Jacobians, eigenvalues and eigenvectors) into more intuitive optimization parameters (learning rates and momentum timescales).

## C  SUPPLEMENTAL METHODS

### C.1  TASKS FOR TRAINING LEARNED OPTIMIZERS

An optimization problem is specified by both the loss function to minimize and the initial parameters. When training a learned optimizer (or tuning baseline optimizers), we sample this loss function and initial condition from a distribution that defines a task. Then, when evaluating an optimizer, we sample new optimization problems from this distribution to form a test set.

The idea is that the learned optimizer will discover useful strategies for optimizing the particular task it was trained on. By studying the properties of optimizers trained across different tasks, we gain insight into how different types of tasks influence the learned algorithms that underlie the operation of the optimizer. This sheds insight on the inductive bias of learned optimizers; i.e. we want to know what properties of tasks affect the resulting learned optimizer and whether those strategies are useful across problem domains.

We train and analyzed learned optimizers on three distinct tasks. In order to train a learned optimizer, for each task, we must repeatedly initialize and run the corresponding optimization problem

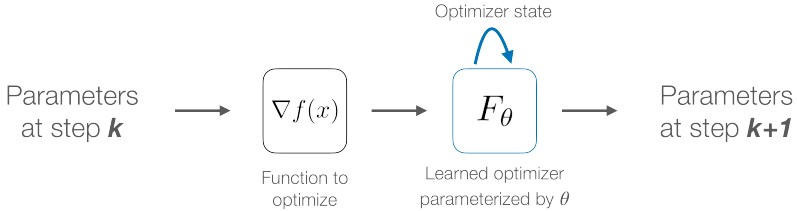

Figure 8: Schematic of a learned optimizer.

(resulting in thousands of optimization runs). Therefore we focused on simple tasks that could be optimized within a couple hundred iterations, but still covered different types of loss surfaces: convex and non-convex functions, over low- and high-dimensional parameter spaces. We also focused on deterministic functions (whose gradients are not stochastic), to reduce variability when training and analyzing optimizers.

## C.2 TRAINING A LEARNED OPTIMIZER

We train learned optimizers that are parameterized by recurrent neural networks (RNNs). In all of the learned optimizers presented here, we use gated recurrent unit (GRU) (Cho et al., 2014) to parameterize the optimizer. This means that the function $F$ in eqn. (1) is the state update function of a GRU, and the optimizer state is the GRU state. In addition, for all of our experiments, we set the readout function $r$ in eqn. (2) to be linear. The parameters of the learned optimizer are now the GRU parameters, and the weights of the linear readout. We meta-learn these parameters through a meta-optimization procedure, described below.

In order to apply a learned optimizer, we sample an optimization problem from our task distribution, and iteratively feed in the current gradient and update the problem parameters, schematized in Figure 8. This iterative application of an optimizer builds an unrolled computational graph, where the number of nodes in the graph is proportional to the number of iterations of optimization (known as the length of the unroll). This is sometimes called the *inner* optimization loop, to contrast it with the *outer* loop that is used to update the optimizer parameters.

In order to train a learned optimizer, we first need to specify a target objective to minimize. In this work, we use the average loss over the unrolled (inner) loop as this meta-objective. In order to minimize the meta-objective, we compute the gradient of the meta-objective with respect to the optimizer weights. We do this by first running an unrolled computational graph, and then using backpropagation through the unrolled graph in order to compute the meta-gradient.

This unrolled procedure is computationally expensive. In order to get a single meta-gradient, we need to initialize, optimize, and then backpropagate back through an entire optimization problem. This is why we focus on small optimization problems, that are fast to train.

Another known difficulty with this kind of meta-optimization arises from the unrolled inner loop. In order to train optimizers on longer unrolled problems, previous studies have *truncated* this inner computational graph, effectively only using pieces of it in order to compute meta-gradients. While this saves computation, it is known that this induces bias in the resulting meta-gradients (Wu et al., 2018; Metz et al., 2019).

To avoid this, we compute and backpropagate through fully unrolled inner computational graphs. This places a limit on the number of steps that we can then run the inner optimization for, in this work, we set this unroll length to 200 for all three tasks. Backpropagation through a single unrolled optimization run gives us a single (stochastic) meta-gradient, when meta-training, we average these over a batch size of 32.

Now that we have a procedure for computing meta-gradients, we can use these to iteratively update parameters of the learned optimizer (the outer loop, also known as meta-optimization). We do this using Adam as the meta-optimizer, with the default hyperparameters (except for the initial learning rate, which was tuned via random search). In addition, we use gradient clipping (with a clip value of five applied to each parameter independently and decay the learning rate exponentially (by a

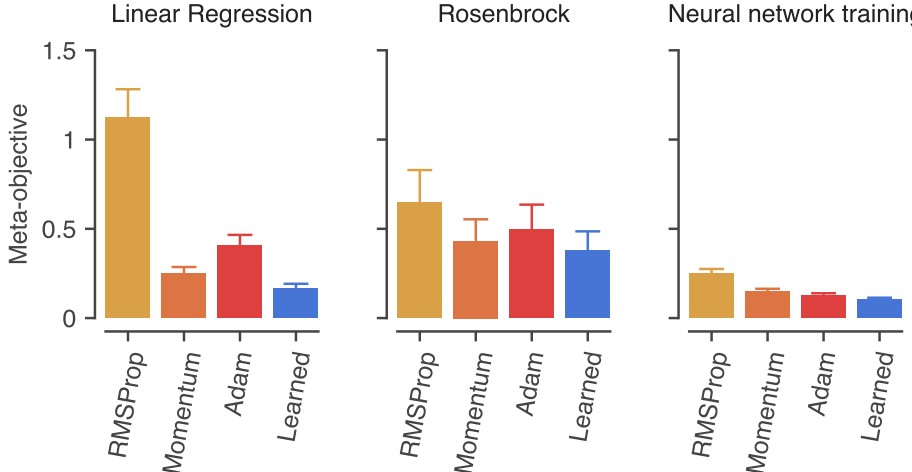

Figure 9: Performance summary. Each panel shows the meta-objective (average training loss) over 64 random test problems for baseline and learned optimizers. Error bars show standard error. The learned optimizer has the lowest (best) meta-objective on each task.

factor of 0.8 every 500 steps) during meta-training. We added a small $\ell_2$-regularization penalty to the parameters of the learned optimizer, with a penalty strength of $10^{-5}$. We trained each learned optimizer for a total of 5000 steps.

For each task, we ended up with a single (best performing) learned optimizer architecture. These are the optimizers that we then analyzed, and form the basis of the results in the main text. The final meta-objective for each learned optimizer and best tuned baselines are compared below in Figure 9.

### C.3 HYPERPARAMETER SELECTION FOR BASELINE OPTIMIZERS

We tuned the hyperparameters of each baseline optimizer, separately for each task. For each combination of optimizer and task, we randomly sampled 2500 hyperparameter combinations from a grid, and selected the best one using the same meta-objective that was used for training the learned optimizer. We ensured that the best parameters did not occur along the edge of any grid.

For momentum, we tuned the learning rate ($\alpha$) and momentum timescale ($\beta$). For RMSProp, we tuned the learning rate ($\alpha$) and learning rate adaptation parameter ($\gamma$). For Adam, we tuned the learning rate ($\alpha$), momentum ($\beta_1$), and learning rate adaptation ($\beta_2$) parameters. The result of these hyperparameter runs are shown in Figures 10 (linear regression), 11 (Rosenbrock), and 12 (two moons classification). In each of these figures, the color scale is the same — purple denotes the optimal hyperparameters.

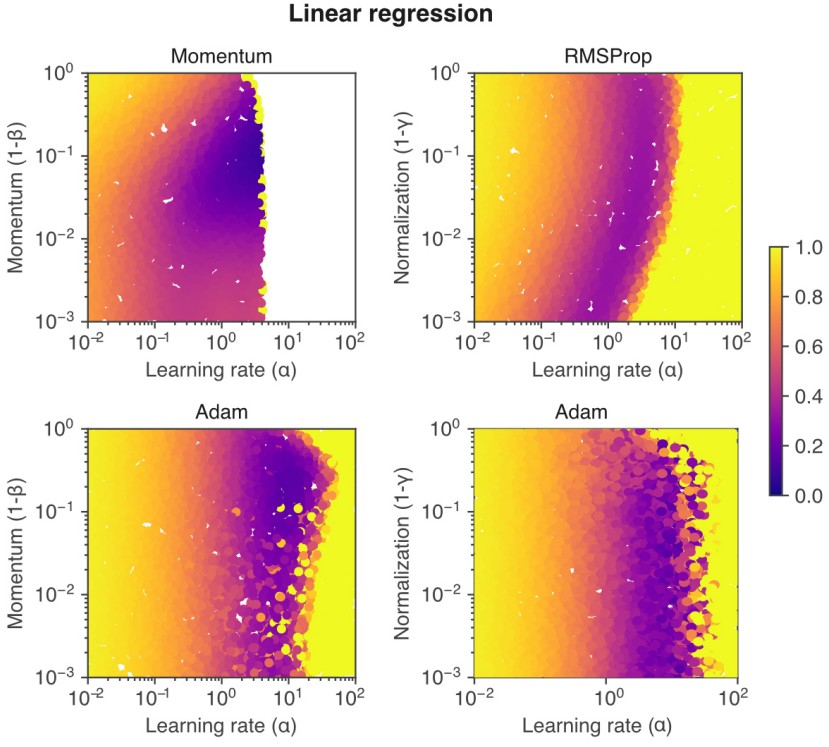

Figure 10: Hyperparameter selection for linear regression.

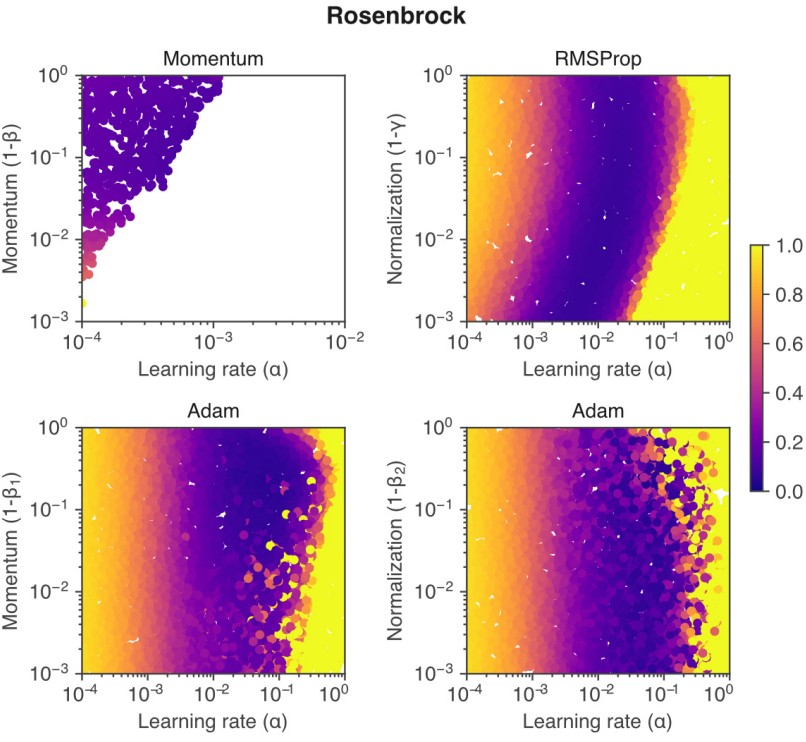

Figure 11: Hyperparameter selection for Rosenbrock.

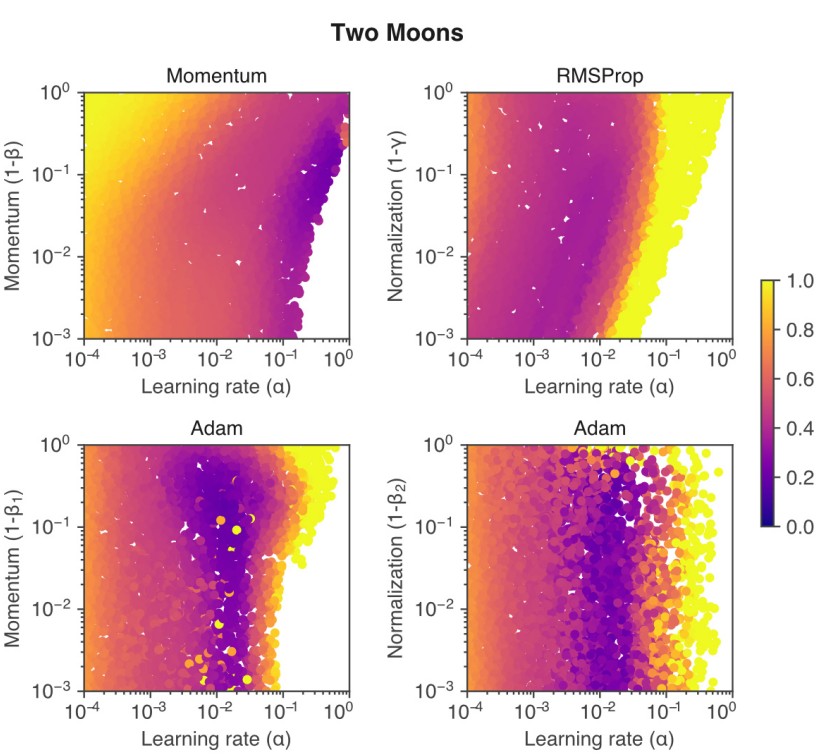

Figure 12: Hyperparameter selection for training a neural network on two moons data.

