# OpenReview forum: "Reverse engineering learned optimizers reveals known and novel mechanisms"
_ICLR.cc/2021/Conference — Reject_

### Official Review · AnonReviewer4 · 2020-10-27
**An intriguing exploration of learned optimizers**

**Rating:** 8
**Confidence:** 4

**Review:**

Hello authors,

Thank you for your submission.  I very much enjoyed reading it.  I found the writing to be clear and only found one grammatical error (detailed below).  As with any black-box system like a learned optimizer, there is naturally a lot of interest in what, actually, the optimizer itself is learning, and why it has learned in the way it has.  In this effort, the authors perform interesting experiments with intriguing results for the first of those two questions.

I found the experiments to be comprehensive and the figures to be adequately described.  It would be great if the authors could provide more details in the main paper on the precise process used to approximate the nonlinear dynamical system.  It could be easy for a reader to assume, without details, that a learned optimizer is "definitely" learning momentum, while missing the nuance of how those results were obtained.  (For the record I am not saying I disagree with the results, only that if more detail can be provided, it would be helpful.)

The experiments are only performed with one form of learned optimizer, using an RNN with 256 GRUs.  What happens if different learned optimizer architectures are used?  Is there reason to believe that they would exhibit similar behavior?  Have the authors perhaps done any experiments with other architectures to see if the results are replicated?  Adding some extra information about this question to Section 4 could improve the paper.

Overall, I think this is a good paper and should definitely be accepted.

Small bits:

 - Can we really drop the subscript notation in (1)?  Couldn't F use other elements of g?  Or are we restricting the situation for this exploration?
 - You might consider switching Figure 2 and Figure 1; the background on learned optimizers meshes well with Figure 2 and so it may be more streamlined to introduce it there.  However, your call.
 - Page 4: "The RNN is is trained" -> "The RNN is trained"
 - It would be really helpful if Figure 3 was on the same page as Section 4.1.  This also applies to Figures 4 and 5.

## Post-rebuttal comments

Thanks again to the authors for the submission.  My concerns are clarified and so I will leave my rating intact.

---

> ### Author Response · Authors · 2020-11-21
> **Response to Reviewer #4**
>
> We have adjusted section 3.2 including more explanation about how the linearization works, with pointers to a full technical writeup in the appendix.
>
> As far as different optimizer architectures, we have trained learned optimizers with different numbers of units (64/128/256). We do not see major differences between these, or across different random seeds used to initialize the training process. This is now mentioned in the text.
>
> Responses to small bits:
> - Yes, we can drop the subscript because we train “per-parameter” learned optimizers that act on every parameter in the target problem in parallel. Thus, the learned optimizer (F), by construction, is only a function of an individual parameter’s gradient.
> - Thanks for the suggestion! We tried swapping Figures 1 and 2, and like it better that way. The figures are swapped in the latest draft.
> - Thank you for the corrections/typos. These are now fixed!
> - We rearranged the figures and text so that they are better aligned.

---

> > ### Comment · AnonReviewer4 · 2020-11-24
> > **Response to Response to Reviewer #4**
> >
> > Thank you for handling these issues.  It's good to know that the results seem to be consistent across somewhat different architectures.  Section 3.2 is clearer to me now.  👍

---

### Official Review · AnonReviewer2 · 2020-10-28
**Toy datasets**

**Rating:** 5
**Confidence:** 3

**Review:**

# Summary

This paper studies learned optimizers and tries to discover what they have learned. The authors argue that the dynamics of the learned optimizers are responsible for those behaviors. Results are mainly presented as visualization and with tools from reverse-engineering dynamical systems.

# Strength

- This seems to be the first paper on re-discovering what the neural optimizers have learned from dynamical system angle.
- Overall, the paper is easy to follow.

# Weakness

- All three experiments are done on toy datasets. Although the authors argue in Section 4 that "These tasks were selected because they are fast to train (particularly important for meta-optimization) and covered a range of loss surfaces (convex and non-convex, low- and high-dimensional)", I completely do not agree. Even in the original neural optimizer paper by Andrychowicz et al. they considered CIFAR (subset). If we only run the optimizers for several hundreds of steps, I think it's totally affordable. It is not convincing that the observed behaviors will generalize to real-world problems.
- In the title and abstract, the authors mention "novel mechanisms". But it is not clear to me which behaviors shown in Section 4 are considered novel? I guess the authors might refer to learning rate adaption, "the effect is to decrease the learning rate of the optimizer when large gradients are encountered". But in Section 4.4, paragraph 4, the authors say ".... similar to the changes observed as the RMSProp state varies," which might suggest this is not novel to human-designed optimizers with a memory mechanism. Even this is novel, this is only observed on three toy datasets.

---

> ### Author Response · Authors · 2020-11-21
> **Response to Reviewer #2**
>
> First, regarding optimization problems. The optimization literature has a long history of studying the behavior of optimizers in simple settings, where the loss function is analytically or numerically tractable. These findings often generalize to more complex problems. For example, optimization theory derived for simple quadratic or convex problems forms the bedrock of our conceptual understanding of the behavior of optimization algorithms on much more sophisticated loss landscapes. In fact, we think the simplicity of these problems is a *strength* of the results, not a weakness. Having careful control over the types of problems being optimized allows us to draw connections between the properties of the learned optimizer and properties of the training task (for example, the connection between condition numbers, momentum timescales, and eigenvalues discussed in Appendix A).
>
> Prior work on training learned optimizers makes compromises when meta-training, in order to accommodate training on larger scale problems. Most common is the use of truncated backpropagation, where the computational graph is only unrolled for a small number of optimization steps. However, this means that the corresponding meta-gradients are biased (Wu et al, https://arxiv.org/abs/1803.02021). By focusing on tasks where we can keep the entire training trajectory in memory, we can avoid these issues--this in our hands results in the overall best learned optimizer, in terms of performance. We focused on these best performing optimizers in this study.
>
> Whether the mechanisms described in this paper also exist in learned optimizers trained on different problems is of course an important question. But we see that as future work that can build on the results presented here. We ask the reviewer to judge this particular work based on if it has improved our collective understanding of what learned optimizers are doing. We think we have revealed many important mechanisms about the behavior of learned optimizers that were previously mysterious. To that end, we think this represents a significant advance in our understanding of learned optimizers, regardless of the fact that this understanding was developed using relatively simple problems.
>
> Second, regarding novelty. We have reworked the section (4.4) that explains the novel mechanism for learning rate adaptation. The overall effect is to decrease the learning rate when large magnitudes are encountered, similar to RMSProp, but the implementation (mechanism) is totally different. In RMSProp or AdaGrad, the learning rate is normalized (divided) by a running average of the gradient magnitude. In the learned optimizer, the effective learning rate varies as a function of the gradient magnitude, by this occurs along changing (input-dependent) fixed points of the optimizer dynamics. The way we think about this is that the dynamics of the system change continuously as the input (gradient) is varied, which corresponds to the S-curve of fixed points. When a particular gradient is encountered for a number of steps, the dynamics of the hidden state will be attracted to the corresponding fixed point. For large magnitude gradients, this corresponds to the tails of the S-curve (away from the final convergence point in the middle). Why have these dynamics? Well, when we look at the update function along the S-curve, we find that the slope of the update function changes--as you move towards the tails, the slope gets smaller. This corresponds to changing the effective learning rate.

---

> > ### Comment · AnonReviewer2 · 2020-11-24
> > **Response to Authors**
> >
> > I read through and appreciate the responses from the authors.
> >
> > It is true that a controlled environment is good for deriving theories, but this does not mean you don't have to test it in realistic settings. For example, it might be interesting to apply the optimizer trained on simple cases to other unseen tasks, as done in 1606.04474. Furthermore, these learned behaviors are mostly known before.
> >
> > As for Section 4.4, even though the behavior is different from RMSProp, it is still not clear this is true and useful for other problems.
> >
> > Therefore, I will keep my original evaluations.

---

### Official Review · AnonReviewer3 · 2020-10-28
**Interesting investigations on the learned optimizer but not enough for a full conference paper**

**Rating:** 5
**Confidence:** 3

**Review:**

Summary:
The author proposed a set of tools to analyze the properties of the learned neural network-based optimizers.

This toolset consists of (i) update function visualizer (ii) linearization of the update equation around the fixed point. The author takes an RNN-based optimizer as an example and analyzes it in the empirical section.

The author investigates several properties including (1) momentum using linearization (2) gradient clipping using the function visualizer (3) learning rate schedule and (4) learning rate adaptation. It shows that the learned optimizer indeed possesses some useful properties.

--------
Review:
Clarity: In general, some parts of the paper is clearly written and motivated such as the problems of the current learned optimizer.  However, the paper quality can be improved if some sections can be more clearer. For example, in section 3.2, the author only mentioned the high-level idea of the proposed tool (linearization) without a detailed mathematical introduction. It would be much more helpful the author can give a brief introduction to the linearization in the Appendix, rather than describe it in words and postpone it in the momentum analysis section.

Novelty: The tool seems to be novel but the idea of linearization is not new, which has been used in other analyses of the dynamical systems as mentioned by the author. As for the function visualizer, it seems to be a plot of the parameter update, which is standard.

Technical soundness:
I have checked some of the details, it seems to be correct.

Significance:
The proposed tools may be useful for analyzing neural network-based optimizers, and help to diagnose their behaviors. The tools are easy to implement and can be applied to a broad class of optimizers.

Weakness:
Although the paper proposed an interesting way to visualize some of the properties of learned optimizers, I think the paper in the current stage is not enough for a full conference paper. Here are some of my concerns and questions for the author:
1. The proposed tools are mainly used to visualize some of the properties of the learned optimizer. This may be helpful for visualizing some properties but it provides no analysis on why it has those behaviors and what advantages these properties when optimizing a function $f$. Or what properties of the function $f$ can induce such behavior of the learned optimizer.
2. The author only demonstrates the usage of the tools in analyzing a single/isolated property. How these properties are combined is not analyzed.
3. The author only analyzes one NN-based optimizer. are there any other forms of the leaned optimizers? What advantages of those optimizers compared to the one used in the paper? Can your analysis confirm those advantages and provide possible reasons? This would be stronger evidence to back up the validity of the proposed method compared to the current analysis without comparisons.
4. As mentioned in the clarity, better use mathematical equations to explain certain terms rather than words. For example, in section 3.1, the slope w.r.t what, the gradient $g$?
5. In section 3.2, I am a bit confused about the difference between the fixed point and convergence point? Any examples of scenarios that it is a fixed point but not a convergence point?
6. In section 4.1 (momentum), for the first figure in the bottom row (Figure 3), is it the eigenvalue point in the regression task? What about the first one used in Figure 7 (Appendix B). Are they the same?
7. In section 4.3, your analysis shows the optimizer has this autonomous behavior. What advantages does this behavior provide? The author also mentions that the parameters should not be updated during the autonomous behavior. How can we confirm this through the plot (Figure 5)?
8. I am not fully sure what you mean in the second paragraph (section 4.4). Do you mean that at the convergence points, you manually give some gradient perturbations, and the fixed points are moved away from the convergence point to get the S-curved shape?

---

> ### Author Response · Authors · 2020-11-21
> **Response to Reviewer #3**
>
> Thanks for the suggestions re: clarity in section 3.2. We have added text and equations that explain section 3.2 in more detail, with corresponding equations.
>
> Responses to additional points:
> 1. It is true that we do not understand why the learned optimizer has those behaviors. However, we think simply presenting what behaviors exist in learned optimizers, and how to identify them, constitutes a significant advance of interest to the larger community. The modern flavor of learned optimizers have been studied since 2016. Since then, there has been no work showing what learned optimizers are doing. Therefore, we think simply showing what learned optimizers are doing is an important first step to understanding why they are doing it.
> 2. Yes, this is pointed out in the discussion, where we mention potential directions for addressing this.
> 3. NN-based learned optimizers form the vast majority of learned optimizers, especially in recent work. Recurrent neural networks were used in the original work by Andrychowicz et al. More recently, fully connected networks have been used (Metz et al), but there haven’t been many direct comparisons of these two architectures in the literature. As we wanted our optimizers to maintain additional state, recurrent networks seemed appropriate. Another variation in the learned optimizer literature comes from the features passed to the optimizer; here we take the simplest approach and only pass in the gradient. Other papers propose methods that pass in additional features (such as momentum velocities), however, we feel that it is a stronger and more interesting result that an optimizer learns to build these features from the raw gradient, as opposed to feeding them in directly.
> 4. Yes, slope wrt. the gradient. This has been clarified in the text, with the accompanying mathematical expression.
> 5. Although we do not see this in any of our trained networks, in theory the recurrent neural network may have additional fixed points in its hidden state dynamics that do not correspond to convergence points of the optimizer. The optimizer has converged when it no longer updates the parameters (\Delta x = 0). This update is a function of the hidden state of the RNN. So, any point in the hidden state space of the RNN that has zero readout can act as a convergence point of the optimizer. However, there may be additional fixed points in the hidden state dynamics that have non-zero readout, meaning that at those locations, the parameters are constantly being updated. This is a bit of an odd scenario, and we don’t see it in practice, but wanted to be explicit about the possibility in the text.
> 6. They are slightly different, and this gets to a subtle point about the optimizer behavior. The eigenvalue in fig 7 (Appendix B) has a momentum timescale that matches the best tuned momentum value. In that optimizer, there were no additional mechanisms found: the optimizer seemed to only do momentum. The optimizer in the main text (Fig 3), trained on the same problem, performed slightly better (had a lower meta-objective) and had additional mechanisms (the learning rate schedule and learning rate adaptation). This more complex optimizer had a different value for the momentum timescale, which we suspect is because it is used in conjunction with the other mechanisms.
> 7. It is not shown in the plot, but that entire trajectory has zero projection on the readout.
> 8. We apologize for the confusion! Each point in the S-curve is an approximate fixed point of the dynamics, the difference is that they are fixed points computed by holding the input (gradient) fixed at different values. When the input is held at zero, we find the final/convergence fixed point (center of the S). As we sweep the gradient from large positive to large negative values, we find different approximate fixed points that make up the S curve. The way to think about these dynamics is that there are different dynamical systems for different input gradients. This means that when the RNN encounters a large gradient, the corresponding dynamics will be drawn to a fixed point away from the convergence point, which induces a change in learning rate. This explanation has been clarified in the text.

---

> > ### Comment · AnonReviewer3 · 2020-11-24
> > **Response to the author**
> >
> > I appreciate the responses from the author. It indeed clarifies some of my concerns. However, I think this is an interesting paper but need more analysis on why the learned optimizer has such behaviours, and maybe any method to improve their performances according to their behaviours. Therefore, I will keep my original evaluations.

---

### Official Review · AnonReviewer1 · 2020-10-28
**Official Blind Review #1**

**Rating:** 5
**Confidence:** 3

**Review:**

This paper aims to shed the light on the work (mechanisms) of the learned optimizers. The main contributions of this paper are the following ones:
 1. The authors found the connection of learned optimization patterns with classical optimization techniques: such as momentum/gradient clipping, etc.
 2. The authors applied the methods from dynamical systems to the analysis of learned optimizers. In particular, the authors used linearization of dynamical systems in vicinity of fixed points to analyse the properties mentioned above, as well as the trajectory of hidden states of learned optimizers during optimization of the inner problem.

Overall, the findings of this paper sound interesting but the paper requires further development. For example, the following directions can be considered:  theoretical analysis of the obtained results, practical applications of the results or investigation of resulting properties on a larger class of optimizers training methods.

To analyse different properties of the learned optimizers the authors considered a set of three simple tasks using an optimizer training method similar to [1] and tried to find the connection between the learned optimization pattern and classical optimization techniques or identify some new behaviour patterns. While the choice of such simple tasks is well-motivated by the fact that optimizers training is very computationally intensive the choice of training method from [1] leads to significant limitations of obtained results from my point of view. In particular, the main difference between [1] and newer methods is its generalization property. As authors from [1] noted the considered method has problems with generalization (for example, if the activation function of the inner task is changed, the learned optimizer is no longer able to generalize and train NN with the different activation function). Due to this it is difficult to say whether findings from this paper can be useful in the more interesting settings or with more advanced optimizer training methods.

Overall, this paper is well-written, but there are some sections of the paper that are difficult to follow. For example, section 3.2. One of the main techniques used in the paper is linearization of dynamical systems around fixed points. I would recommend adding into the paper a brief description of the method with its limitations and all necessary definitions. In particular, I would provide the rigorous definitions of fixed point/stable fixed point/global fixed point of dynamical systems.

Some other questions and issues:
1. Appendix C.2 has incorrect reference to Figure 5.
2. Figures 10/11/12. Adam optimizer has 3 parameters alpha, beta, gamma. The plots corresponding to Adam have 2 parameters each, what is the value of the third parameter?
3. The authors of [2] noticed that it is beneficial to provide additional input features that are borrowed from classical optimization methods (e.g. momentums) besides gradients.  Could you please clarify how these observations align with the findings that learned optimizers are able to learn classical techniques?
4. How will the method properties change if the hidden size of the learned optimizer will be decreased (e.g. from 256 to 40/20)?

[1] Marcin Andrychowicz, Misha Denil, Sergio Gomez, Matthew W Hoffman, David Pfau, Tom Schaul, Brendan Shillingford, and Nando De Freitas. Learning to learn by gradient descent by gradient descent. In Advances in neural information processing systems, pp. 3981–3989, 2016.

[2] Olga Wichrowska, Niru Maheswaranathan, Matthew W Hoffman, Sergio Gomez Colmenarejo, Misha Denil, Nando de Freitas, and Jascha Sohl-Dickstein. Learned optimizers that scale and generalize. arXiv preprint arXiv:1703.04813, 2017.

---

> ### Author Response · Authors · 2020-11-21
> **Response to Reviewer #1**
>
> We agree with the reviewer that understanding how and when learned optimizers generalize is an important question. In fact, we think this paper presents a first step towards goal. We believe that in order to deeply understand how learned optimizers generalize, we need to first understand want they are doing. For example, our analysis reveals not only the types of mechanisms within learned optimizers (momentum, etc) but also quantitative properties of those mechanisms (eg. the momentum timescale). We can then relate these quantities back to properties of the training distribution (for example, in Appendix A we show how the learned momentum timescale is related to the timescale predicted from the condition numbers of the Hessian of the training task. This allows us to identify quantitative properties of known mechanisms of learned optimizers. Future work building on these results could then look at how these properties change depending on the training tasks, getting at questions of generalization.
>
> As part of this first effort, we also made the conscious decision to focus on optimizers trained on a narrow set of well defined optimization tasks. The “advanced optimizer training methods” largely consist of training learned optimizers on large sets of tasks. The way we see it, we first need to understand how learned optimizers work when trained on a specific task. Then, we can see how they change when trained on multiple tasks, or large sets of tasks. This paper addresses the former.
>
> We agree that understanding more complex learned optimizers, trained on for a long time on large task distributions, is of interest. However, this training procedure is cumbersome, and the resulting learned optimizers often do not work well across the entire training distribution of tasks (e.g. Metz et al 2020, https://arxiv.org/abs/2009.11243). We think the tools and analysis presented in this work lay the foundation for future work to explore these questions.
>
> To date, there has been little to no work on understanding what learned optimizers parameterized by neural networks are doing. The work presented here was a significant advance in our own understanding of how learned optimizers work, and thus we made the decision to write up these results.

---

> > ### Author Response · Authors · 2020-11-21
> > **Response to Reviewer #1 (continued)**
> >
> > Thanks for the suggestions re: clarity in section 3.2 We have added text and equations that explain section 3.2 in more detail, with corresponding equations.
> >
> > Responses to other questions:
> > 1. We have fixed the incorrect reference, thanks.
> > 2. Each pairwise scatter plot shows points for all values of the third parameter. That is, we take the randomly sampled values (which exist in 3D) and project them onto 2D slices to form the visualizations in Fig 10-12.
> > 3. We suspect this is due to difficulties in training learned optimizers. The authors of [2] used truncated (partial) unrolls for training learned optimizers, which is known to suffer from truncation bias. In addition, they trained on a large set of problems, which introduces stochasticity in the resulting gradients. These make training learned optimizers difficult, so perhaps directly providing certain signals (such as momentum variables) made training easier. However, we think it is a much stronger statement to say that a learned optimizer recovers momentum *only when fed gradients*, rather than to say that one uses momentum when provided momentum variables.
> > 4. We trained learned optimizers with different numbers of GRU units (64/128/256) and did not see large differences in performance, or in behavior. This is now mentioned in the paper.

---

> > > ### Comment · AnonReviewer1 · 2020-11-24
> > > **Response to the authors**
> > >
> > > I would like to thank the authors for their reply. In the revised version the authors added more details into section 3.2 that significantly improved clarity of the paper. I carefully read the authors’ feedback and other reviews. I still believe that obtained contribution is rather proof of concept. The authors observed some patterns in learned optimizers and it is interesting and promising. However, the claim that observed mechanisms in the learned optimizers are responsible for their superior performance is debatable. There are no experiments that show that found patterns are solely responsible for the performance of learned optimizers. I think it would be more precise to say that authors tried to find similarity between learned optimizers and some well-known techniques. Since the authors did not provide any significant additional experimental results, I would leave my assessment unchanged.

---

### Author Response · Authors · 2020-11-21
**General response to all reviewers**

We wish to thank all of the reviewers for great feedback and suggestions on how to improve the paper! We have clarified the text and explanations in sections 3.2 (introducing dynamical systems analyses) and 4.4 (explaining how the learning rate adaptation mechanism works). In addition, we have reorganized some sections and added additional references.

We hope all of the reviewers reconsider their ratings in light of our responses. From our perspective, there has been little to no understanding of what learned optimizers of any form are doing, when trained on any task. The results presented here constitute a significant advance in our own understanding of what learned optimizers are doing, and we strongly believe this work will be of interest to the greater community as well.

Thanks again for your time and attention!

---

### Decision · Program_Chairs · 2021-01-07
**Final Decision**

**Decision:**

Reject

**Comment:**

A line of work since 2016 has investigated learning NN-based optimisers, which produce optimisation updates by processing loss/gradient info with neural networks. This paper tries to understand the learned dynamics of these NN-based optimisers by linear approximation to the learned non-linear dynamics. Visualisation of these approximations are shown on 3 optimisation problems: linear regression, Rosenbrock function, and a toy neural network classification problem, with the hope of covering different types of objective landscapes.

Reviewers agreed that the paper studies an important research question, which would interest researchers working on meta-learning learning algorithms. However there are several major concerns raised by the reviewers: (1) the example optimisation problems are toyish, and (2) the paper does not explain very well the link between the visualised behaviour and the better optimisation results, i.e. it is unclear to the reviewers why the learned dynamics lead to better optimisation results.

While I am not too concerned about issue (1), I think issue (2) is a significant one, flagging that the clarity of the paper needs to be improved. Ultimately, the paper is motivated by the question "How is a learned optimizer able to outperform a well tuned baseline?", so a reader would expect some clear explanation towards answering this question. Also some reviewers are concerned about the fact that only the RNN-based optimiser in Andrychowicz et al. (2016) is analysed; since there exists other forms of learned optimisers, focusing on studying only one type of them might lead to early conclusions that are not so accurate.